# Physical Education: At the Centre of Physical Activity Promotion in Schools

**DOI:** 10.3390/ijerph20116033

**Published:** 2023-06-02

**Authors:** Lorraine Cale

**Affiliations:** School of Sport, Exercise and Health Sciences, Loughborough University, Loughborough LE11 3TU, UK; l.a.cale@lboro.ac.uk

**Keywords:** physical activity promotion, physical education, physical education teachers, whole school approaches, young people

## Abstract

Whilst recognising and advocating for the role and importance of schools and whole school approaches to the promotion of physical activity in schools, this paper argues that physical education (PE) should be at the centre of and driving schools’ efforts to promote physical activity. Various reasons are given for this, with these broadly centring on the unique goal, nature, and responsibilities of the subject with respect to promoting physically active lifestyles and health-related learning. Furthermore, there have been positive strides in recent years to support this endeavour and that serve to highlight, strengthen, and reinforce the focus and responsibility of PE in the promotion of physical activity. In light of these, it is suggested that it is a pivotal time for PE. Equally, it is accepted that PE faces some longstanding challenges that are hindering and raise questions concerning the subject’s physical activity promotion efforts. Despite this, it is contended that these should not be unsurmountable, and more recent developments should also help the subject to realise its physical activity promoting potential moving forwards. In particular, the critical importance of high-quality PE that has young people at the core is highlighted. It is concluded that it is both time and timely for the PE profession to be bold, have confidence, and grasp these opportunities and ensure that high-quality PE is central to the explicit planning and co-ordination of meaningful, coherent, relevant, and sustainable physical activity opportunities for young people in schools.

## 1. Introduction

It has been a real privilege to have served as the Guest Editor for this Special Issue on ‘Promoting Physical Activity in and Through Schools’ and to have read so much of the excellent research and wide-ranging work being undertaken globally in this important area. I also welcome this opportunity to contribute a short opinion piece to the Issue. In doing so, I should declare my own background and personal interests up front, as these have inevitably shaped my own research, practice, and thinking over the years, and they explain the stance I take in this paper. I am a former physical education (PE) teacher and current PE researcher and teacher educator of over 25 years with a particular and longstanding interest in health and health-related learning within PE. During the course of my career, I have also delivered and developed professional development courses and seminars for PE teachers and other practitioners across the UK and produced a number of school-based curriculum resources and practical texts to support schools and teachers in this area of their work. It goes without saying, therefore, that when I discuss PE and the PE profession’s contribution to physical activity promotion in schools in this paper, I see myself very much as part of the profession with a responsibility and key part to play. 

Of course, the important role and contribution PE makes to promoting health and physical activity has long been acknowledged by the profession and in the literature, see for example, [1,2,3,4]. In this paper, though, I argue that PE needs to be at the very centre of schools’ efforts to promote physical activity. Indeed, health, and the promotion of healthy, active lifestyles specifically, represents a key component of the aims or purpose of the PE curricula in many countries including, for example, in Scotland, Wales, Ireland, England, United States, Australia, New Zealand, Sweden, and Norway to name but a few. Furthermore, governments, organisations, and agencies worldwide have and continue to recognise the crucial role of PE in promoting health and physical activity through various strategies, initiatives, and reports. For instance, the importance of quality PE in enhancing physical activity participation is recognised in the World Health Organization’s (WHO) Global Physical Activity Action Plan ‘More Active People for a Healthier World’ [5], with the Plan proposing a number of actions relating to PE. Similarly, UNSECO [6] (p. 20) sees the outcome of quality PE to be ‘a physically literate young person, who has the skills, confidence, and understanding to continue participation in physical activity throughout their life-course.’ Furthermore, given the negative impact of the pandemic on young people’s health and physical activity [7,8,9], the role of PE would seem to be more critical than ever. Consequently, this led to calls by the Association for Physical Education (afPE), the representative subject association in the UK, for the subject to be a key part of the recovery and levelling-up agenda, ‘providing the springboard to a life of physical activity and sport’ and ‘giving opportunities for all to thrive—and not just survive—in a challenging world’ [10] (pp. 6–7).

## 2. The Rationale

The rationale for putting PE at the centre of schools’ efforts to promote physical activity may seem obvious and some may question the need to even argue the case. Aside from the above, first and foremost, schools are learning environments, and their primary (though not only) concern is students’ learning. At the centre of learning, lies the curriculum and learning is also fundamental to a lifelong participation in physical activity [11,12,13]. PE is, furthermore, the only curriculum subject whose primary focus is on the body and the development of physical competence and physical literacy, and which strives to ensure that children can (and know) how to move efficiently, effectively, and safely [14]. For many youngsters, it may therefore be the only context within which they will learn and develop the skills, knowledge, confidence, values, and attitudes necessary to be able to enjoy and reap the benefits of a physically active lifestyle. Moreover, whilst physically active learning (i.e., the integration of physical activity into lessons in areas other than PE) is growing in popularity [15,16], the dual focus of PE on ‘learning to move and moving to learn’ provides a truly authentic and meaningful context for ‘moving’ for young people. As a profession, PE and the PE workforce also have, or should have, the appropriate knowledge, expertise, enthusiasm, and commitment to be role models and effective champions of and promoters of physical activity. For example, most recruits to the PE profession are physically active, fit, individuals [17]. Thus, the argument for PE’s central role in promoting physical activity is logical and compelling, at least in theory. However, in practice, PE faces various longstanding challenges and criticisms that hinder and even cast doubt over the subject’s role, efforts, and ability to effectively promote physical activity. Some of these ‘hindrances’ are highlighted later and suggest that PE’s position in this regard can not be taken for granted.

As alluded to, the role of PE in health should be one of physical, health, and physical activity ‘education’ (i.e., enhancing learning) and go beyond that of purely promoting physical activity via the provision and facilitation of physical activity opportunities. Such opportunities can, and should, I contest be integrated alongside and in addition to PE within and beyond other areas of the curriculum and clearly serve a purpose. In isolation though, they are undoubtedly limited and likely to have minimal impact. It is within and through PE where explicit learning experiences need to be planned for young people to equip them with the health-related knowledge, understanding, skills, and attributes required to be physically active for life and to make critically informed decisions concerning their lifestyles and behaviours. To achieve the above, a broad and holistic knowledge and understanding and range of physical activity experiences are clearly required [11]. To try to illustrate the breadth and scope of learning considered fundamental to the successful engagement in lifelong physical activity, specific health-related learning outcomes have been proposed. These include cognitive, affective, and behavioural outcomes and span four categories (safety issues, exercise effects, health benefits, and activity promotion) and age ranges (4–7 years, 7–11 years, 11–14 years, and 14–16 years) [11]. By explicitly planning for and addressing health-related learning in this way, physical activity promotion becomes fully, comprehensively, and progressively embedded within the subject and is likely to be more effective. Based on the planned and taught curriculum, PE teachers should then be able to identify, co-ordinate, and support other school-wide activities and opportunities (both within and beyond other curriculum areas) to complement, build on, and reinforce the learning and physical activity experiences provided in PE, and thereby ensure a coherent, meaningful, relevant, and attractive physical activity offer and experience for young people. In essence, then, PE should be the bedrock and provide the springboard to schools’ further and whole school physical activity promotion efforts. Without such central involvement, some otherwise potentially valuable wider school efforts arguably run the risk of being bolted on, ad hoc, and piecemeal, and therefore less or even ineffective.

Some positive strides have been made in PE in recent years to support the above endeavour and that serve to highlight, strengthen, and reinforce the focus and responsibility of PE in the promotion of physical activity within and beyond the subject. For example, the Promoting Active Lifestyles (PAL) Project [18], represents a flexible, principle-based approach to supporting PE teachers in becoming more effective champions of physical activity. The PAL project involved the co-construction (with PE teachers and trainee teachers) of 20 principles associated with the promotion of active lifestyles, 10 of which were PE-specific and 10 of which were whole school principles. Alongside principles that focus on enhancing students’ health-related learning, a number of the PE-specific principles focus on encouraging teachers to reflect on and make minor changes to some standard pedagogical practices within PE that hinder active learning time. Examples include the following: limiting/reducing changing time, maximising active learning time during lessons by limiting/reducing the time spent giving instructions and queueing/waiting to access equipment/resources, moving students on to the next task without stopping the whole class, where appropriate, and including assessment of learning and progress in active ways (e.g., show me; demonstrate; shadow) [19].

A further development which recognises the critical role of PE in the promotion of physically active lifestyles is the Health-based PE (HbPE) pedagogical model that was first developed by Haerens et al. [20]. The model has as its central theme ‘valuing a physically active life’ [20], the teaching of which involves ‘helping individuals to see the intrinsic benefits of physical activity and securing meaningful experiences through their PE journey’ [21] (p. 61). Bowler and Sammon [21] highlighted four learning aspirations to help realise the main idea of valuing a physically active life and that aim to support all young people to become habitual, informed, motivated, and critical movers. The authors, furthermore, proposed four critical elements of the model that reinforce the central role of PE and PE teachers in positively influencing young people’s physical activity levels: (1) the teacher promotes meaningful physical activity; (2) the teacher supports students to be informed movers; (3) the teacher creates a needs-supportive learning environment; and (4) the teacher encourages students to become critical movers.

As already touched upon, there is much scope for other curriculum areas and avenues of the school to make important contributions to the promotion of healthy, active lifestyles through whole school approaches to the promotion of physical activity. Examples of such approaches include the Comprehensive School Physical Activity Program (CSPAP) [22], Health Optimizing PE (HOPE) [23] and the Creating Active Schools (CAS) framework [15]. It is of course, no coincidence that PE is a central component of such approaches. The CSPAP is a multi-component framework that includes five components: (1) quality PE; (2) physical activity during the school day; (3) physical activity before and after school; (4) staff engagement; and (5) family and community involvement [22]. Quality PE is recognised to be the core component of an effective CSPAP, and it is suggested that once schools have established this, the next step should be to focus on implementing one other component well, and so on. Furthermore, to support the implementation and sustainability of a successful CSPAP, an initial recommended step is to elect a Physical Activity Leader, which it is anticipated would most likely be a PE teacher [24].

Health Optimizing PE (HOPE) [23] is similar to CSPAP in that it too is a comprehensive, whole school physical activity programme. This approach focuses on teaching and learning about health and developing the knowledge and skills for lifelong physical activity participation in and through PE, alongside a variety of other contexts or strands. For example, via before and after school physical activity, community-based physical activity, and the integration of HOPE across other subjects. Meanwhile, the Creating Active Schools (CAS) framework [15] is another multiple component approach that aims to embed physical activity at the heart of school life and support schools to increase their students’ daily physical activity. PE is one of seven opportunities or contexts within the framework for influencing physical activity alongside other curricular lessons, break/lunchtimes (recess), trips and events, active travel, and family/community physical activity out of school [15]. Co-production with a range of key stakeholders (e.g., school leaders; teachers; other school staff; children/young people; and parents/guardians) is a particularly attractive feature of the CAS approach, with schools encouraged to collectively design bespoke physical activity programmes to reflect their specific contexts. 

Lastly, it is relevant and worth acknowledging the work and outcomes to date of a Taskforce on ‘The Future of PE: The Heart of School Life’ recently set up by the Association for Physical Education (afPE) in the UK. In 2021, and in recognition of the contribution of PE to a healthy, active lifestyle and young people’s health and wellbeing, the Taskforce published a report recommending for there to be an urgent review by central government of the status of PE as a foundation subject within the National Curriculum for England. The report further recommended that PE be classified as a core subject of the National Curriculum (alongside the other core subjects of Mathematics, English, and Science) [10]. (Curriculum subjects within the National Curriculum in England are divided into two categories, core subjects and foundation subjects. Both are compulsory, but the core subjects have greater levels of prescription and detail regarding what should be delivered, and hence are typically afforded more curriculum time, status, resource, and enhanced initial teacher training and professional development opportunities ([25]). The aspiration for PE to become a core subject was also echoed in a recent House of Lords Committee National Plan for Sport, Health and Wellbeing, which additionally called for there to be ‘greater emphasis on physical literacy and making PE and school sport a fun, enjoyable and inclusive experience’ [26] (p. 5). The work of the afPE Taskforce was initially triggered by the impact of COVID-19 during which time it was argued that the subject needed to ‘be at the heart of the response’ and our recovery, but beyond this ‘at the heart of school life on an ongoing basis’ [10] (p. 200). For PE to become a core subject, the report highlighted how, alongside other requisites, high-quality PE delivered by highly skilled teachers needed to be prioritised, and increased PE time and more coherent links between PE and the physical and mental health and wellbeing of young people were required.

The Taskforce’s report [10] was subsequently considered by government and provided an impetus for afPE to launch an 18-month action-research pilot project to review the impact of PE as a core subject on the school curriculum in England. The pilot comprises three phases and is ongoing, but evident in the vision for PE as a core subject to date are expectations across three key areas including the PE curriculum, school sport programme, and physical activity provision [27]. Thus, alongside delivering high-quality PE, the need for the subject to take responsibility for driving school sport and broader physical activity provision within the school is recognised. The report from the second phase of the pilot, which explored the perspectives of key stakeholders on PE as a core subject, has just been published [28]; it makes five key recommendations to inform the next phase of the research and support the advancement of PE as a core subject agenda. These include the need to outline a clear vision for PE as a core subject, future-proof the subject, continue to develop the profession, promote the potential of PE to the public at large, and secure commitment from wider stakeholders to protect and empower PE [28]. Whilst this work is still in progress, the next couple of years will no doubt be a critical time for PE in England. Regardless of the outcome, the resulting research and debate should certainly raise the profile, status, evidence base, and understanding of the subject amongst government, the public, and other stakeholders and of its centrality in promoting healthy active lifestyles and impacting the health, physical activity, and wellbeing of young people. It is also hoped that this work will have benefits for PE elsewhere.

## 3. Hindering PE’s Potential

As mentioned earlier, various longstanding challenges face PE that hinder the subject’s efforts in effectively promoting physical activity. Such hindrances have been acknowledged in the above work as well as in the wider literature. For instance, PE, and particularly health within PE, reportedly has a low or marginal status relative to other subjects or aspects of the curriculum [2,10,29,30,31]. With respect to health within PE, the aims and purposes of the subject have and continue to be widely debated and contested, see for example, [32,33], with health and physical activity outcomes representing just one goal amongst many (such as the development of physical and social skills, moral values/citizenship, spirituality, intellectual ability, and recreation). Indeed, the ‘proper role for PE in health’ [1] (p. 209) has been the topic of discussion for some time with critics questioning whether PE should have a public health goal, see [34,35]. This scepticism stems from the uncritical, simplistic, and narrow way in which PE teachers have been found to engage with the area, which in turn manifests in their curricula and pedagogies, see [36,37,38,39,40]. This point and some of the reasons for this are discussed further on.

Thus, whilst I and many others consider a key goal of PE to be the promotion of healthy, active lifestyles, it should be acknowledged that some favour other outcomes or in fact remain unconvinced. Even as a firm proponent myself, I am mindful of the limitations and boundaries in terms of what the subject can and can not achieve and what it should and should not be held accountable for [30]. For example, PE cannot meet all of young people’s physical activity and health needs [2,41], be responsible for young people’s health outcomes [1], nor can it solve societal health problems such as ‘…obesity, physical inactivity, drug misuse… youth suicide, etc.’ [42] (p. 202). That said, PE can and should stimulate interest, enjoyment, knowledge, competence, and expertise in physical activity and sport for health and wellbeing [41] amongst all young people, which, first and foremost, needs to be through the delivery of high-quality PE.

On a different but related note, the afPE Taskforce [10] suggests that the unhelpful tendency for PE to be conflated with school sport and physical activity hinders the ability of the PE workforce to deliver its full potential. Surely the PE profession should be central to educating all stakeholders with regards to the differences, delineating the respective roles, contributions, and relationships between different disciplines and groups and, as noted earlier, to co-ordinating and supporting the collective (and high-quality) PE, sport, and physical activity offer within schools. Whilst this too may seem obvious, it may not be so straightforward in practice in that such unhelpful conflations and views regarding what PE is and should be are seemingly not only prevalent amongst government, politicians, and the public but are often found and reinforced within the profession, see for example, [43]. Reflecting on the importance of PE, Lawson [17] (p. 42) refers to how ‘competing constituencies with diverse goals’ (including sport, health, and education) complicate matters and claims that whilst relations between them can be developed, they rarely are. 

A further and longstanding hindrance to PE includes the continued dominance of the traditional multi-activity model to PE [33]. This is a sport-focused model that involves prioritising the teaching of activity content, techniques, and skills typically through teacher-directed approaches. The privileging of this approach is said to stem from PE teachers’ sport and performance-oriented backgrounds and philosophies [44,45], but it has been widely criticized for being narrow, unappealing to many young people, and therefore counterproductive to the promotion of a healthy, active lifestyle [29,33,45,46]. Perhaps not surprisingly, this approach is then reportedly replicated in PE teachers’ health-related teaching and practices [47,48]. For example, with the teaching of health often focusing on a limited and narrow diet of ‘fitness’, ‘performance’, and related activities [29,30,49,50] at the expense of a broader range of health-related lifestyle activities. Similarly, teachers’ interpretations and conceptualisations of health and, consequently, the associated health-related learning that they seek to impart have been decried for being minimal, uncritical, narrow, and simplistic and based on limited and limiting discourses of health [30,38,47,48]. Namely, discourses of health focussed on risk, ill health, the body, normality, and individual responsibility (healthism), which position health as a moral and personal responsibility [44,47] rather than focussing on more sociocultural, ecological, and critical perspectives to health and on developing the resources to lead a physically active life.

Thus, it is important to continue to challenge and question traditional PE practice in terms of its compatibility with health-related learning goals, behaviours, and with the effective promotion of physical activity specifically. PE needs to place value on and incorporate a broad range of ‘purposeful’ physical activities and health-related aspects of learning into the curriculum [30], thereby extending the vision and ultimately the potential for the subject. As appropriate to the school context, activities might include lifestyle sports, play sports and competitive sports, dance, outdoor and adventurous activities, and individual and new health-focussed activities (e.g., cycling, scooting, golf, yoga, Pilates, mindfulness activities, paddle boarding, surfing, and kayaking). Meanwhile, learning about health needs to be critical and holistic and cover mental, social, emotional and physical health and wellbeing as well as lifestyle management skills.

The above issues may be explained and further compounded by limitations in PE teachers’ knowledge and understanding of health and a lack of relevant initial or continuing professional development in the area [1,29,31]. Indeed, these and other issues have led to some doubts over and pertinent questions being raised concerning the extent to which many PE teachers are adequately trained, equipped, and, therefore, best placed to effectively promote and/or assume responsibility for the promotion of physical activity [29,31]. These same authors have also cited a bias towards sport in many PE teachers’ professional development profiles and in the sources of health knowledge they rely on. Health and health behaviours, including physical activity, are complex, with numerous personal, social, cultural, and environmental factors and inequalities within and beyond PE (and schools) influencing young people’s health and physical activity [39,51]. The assumption, therefore, that PE teachers have the knowledge and understanding to address these factors in appropriate and meaningful ways with their students with limited or no relevant training would seem to be naive.

For the reasons already outlined, and while fully committed to the notion of PE being at the centre of schools’ efforts to promote physical activity, I am still of the view that PE should not hold sole responsibility and that there needs to be a broader approach and a commitment to whole school approaches [31]. The profession has a duty to ensure that PE practitioners are, and remain, fit for purpose [1], which may require some upskilling so that they understand health, physical activity, and physical activity promotion and their complexities and are able to respond to the physical activity and health needs and interests of today’s youth. Clearly, teacher education has a key role to play in adequately preparing teachers in this area from the outset, with this initial preparation being further developed and updated through relevant continuing professional development. With regards to the former and in response to the aforementioned concerns and challenges, a position statement on health-related PE (HRPE) in initial teacher education (ITE) was produced by Harris and Cale [52]. This followed a PE ITE conference in March 2020, during which delegates were invited to reflect on and review the health-related component of their own PE ITE programmes, move towards research-informed, evidence-based practice in this area, and contribute to the development of the position statement. Some key recommendations are made within the statement, such as, for PE teacher educators to include a specific HRPE component within their PE ITTE programmes, which is kept under regular review; incorporate minimum content requirements such as clarification of key concepts, appropriate learning outcomes, and understanding the range of physical, mental, and social health benefits of physical activity; and to adopt a critical approach to the area of work and encourage others (schools, teachers, pupils) to do likewise, and ensure that their health-related PE philosophies, policies, practices, and pedagogies all align [52].

## 4. Realising PE’s Potential

Whilst the challenges are real, and the ‘health landscape’ in PE is clearly complex [47] (p. 1163), they should not be unsurmountable. Indeed, the case and opportunities for PE to be at the centre of schools’ efforts to promote physical activity are currently stronger than ever and there are many new and positive developments that, if embraced, will help PE to realise its position and potential in this regard. This view is further echoed by Jess, McMillan, and Carse [53] who propose how a number of recent developments and conditions offer a positive and ‘bright’ future for PE and its relationship with health. 

Firstly, in the UK and in light of the work of the afPE Taskforce, the ‘PE as a core subject’ debate has been comprehensively aired and given serious consideration by the government. It is essential to retain the momentum of this work and continue to develop the evidence base to support the recommendation for PE to be elevated to core subject status. Extending the work and debate to other countries could also help the collective cause of raising the status of PE such that its potential can be fully realised. In their report, the afPE Taskforce identified a number of opportunities in the external environment now serving to support PE such as greater recognition by government of the need to be active, acknowledgment by headteachers and parents that physical and emotional wellbeing should be a priority for schools, and more willingness across the PE, sport, and physical activity sector to work together [10]. Alongside their headline recommendations, the Taskforce also made a number of further recommendations that concur with many of the points raised earlier and that would address some of the challenges identified. For example, for government and the sector to prioritise high-quality teaching of PE; establish better, more coherent linkages between PE and the health and wellbeing of young people; for teacher training to be more comprehensive for non-specialist staff; for there to be a wider and deeper ‘training’ curriculum to allow all staff to develop greater skills and knowledge; for there to be more systematic approaches to continuing professional development; for the workforce to become more skilled and confident in articulating the impact of PE on young people and the life of their school; and for the profession to consider offering greater breadth in the PE curriculum [10] (pp. 22–23).

Also critical to making strides in this area is further advancing our understanding of effective pedagogies in relation to the teaching of health in PE [1], and there have been some promising developments in this respect and upon which to build. Two such pedagogical approaches were highlighted earlier, the PAL project and the HbPE model, both of which have resulted in some encouraging outcomes for teachers and students including changes in teachers’ pedagogies and increased physical activity levels and positive responses from students [18,54]. Of additional relevance in fostering healthy, active lifestyles in and through PE are the affective domain and pedagogies of affect [55,56], which involve explicitly building young people’s affective attributes including confidence, motivation, determination, self-esteem, and resilience through PE. It is pleasing to see more attention being given to these in both research and practice, which suggests an important shift towards a broader and more holistic approach to learning and the subject.

Other pedagogical developments and approaches that are attracting increased interest and have merit include student-driven, participatory pedagogies [57] and activist approaches [58] that see a shift in power between teacher and students, as well as strengths-based approaches [42] that recognise young people’s strengths and interests and encourage their agency. These approaches place young people at the centre and typically involve and empower them in designing and influencing their curriculum, learning, and physical activity opportunities [47,58,59]. Indeed, young people and young people’s experiences being at the core of our efforts is central to the principles of good, effective pedagogy [30] and to ensuring that teachers’ efforts are aligned with the needs and interests of their students.

Returning to the ‘bright future’, Jess and colleagues [53] view PE and health to have (noted earlier), they claim a new set of conditions are currently helping to both integrate PE and health and re-position the subject as a more central feature within education. These conditions include the following: (i) a shift in thinking (to move beyond a simple ‘one-size-fits-all’ approach to see different ways of thinking and practising come together across education, PE, and health); (ii) holistic approaches to PE (such as meaningful PE, strengths-based learning, and the HbPE model highlighted previously that focus not only on physical learning but also on the cognitive, social, and emotional domains); (iii) holistic approaches to health education (moving beyond the focus on physical disease towards more holistic, ecological, and social-ecological perspectives); (iv) holistic benefits of physical activity (recognising the social, emotional, and mental benefits associated with individual lifestyle, health, and wellbeing, alongside the physical); and (v) more stakeholders (such as politicians, policy-makers, national organisations, school leaders, health professionals, sport coaches, parents/carers, the media, and the public) now more involved in the development of PE. 

The shift to more holistic PE curricula and approaches, including those with respect to health within PE, has been noted earlier and elsewhere [30,47]. Gray and colleagues [47], for example, highlight how, with the exception of the curriculum in England, recent curriculum developments in the UK (i.e., in Northern Ireland, Scotland, and Wales) conceptualise health and wellbeing holistically, taking into account social, emotional, and mental as well as physical wellbeing. (Each devolved government within the four home nations of the UK (England, Northern Ireland, Scotland, and Wales) has the responsibility for developing and delivering their own education system. There are therefore significant differences in PE policy and in the structure and content of the National Curricula across the four countries.) Gray et al. furthermore note how the curricula in these home countries are often associated with developing skills and capacities to enhance health and wellbeing as opposed to avoiding risks to protect health and wellbeing [47]. These developments furthermore seem to have followed the lead of other countries such as Australia and New Zealand. Australia’s ‘futures-oriented’ curriculum is underpinned by five interrelated propositions that include a focus on educative outcomes; adopting a strengths-based approach; the development of health literacy skills; valuing learning in, about, and through movement; and a critical inquiry-based approach [60]. Meanwhile, New Zealand’s Health and Physical Education curriculum focuses on wellbeing, with four underlying concepts guiding learning including attitudes and values, Hauora (a Maori philosophy of wellbeing), health promotion, and a socio-ecological perspective [61].

With regards to Jess and colleagues’ [53] fifth condition, some stakeholder involvement in PE such as the outsourcing of the subject to sport coaches and commercial providers has faced criticism [62,63,64], not least because it can lead to different sectors competing for space and resource [53]. However, others see the benefits to be gained from increased stakeholder involvement. Carse, Jess, and Keay [65] contend that this stops PE from being a ‘closed shop’ and expands the network of parties who can potentially work together to develop a more coherent and connected educational vision for the subject. Thus, it would seem the profession needs to embrace and work with key stakeholders as allies for the collective good of both the subject and more importantly our young people and their physical activity experiences and opportunities. The critical importance of working with and enlisting the support of stakeholders has been reinforced in the recent ‘PE as a core subject’ debate and pilot research [28], and I would argue PE needs to be at the centre of and driving such collaborative and joined-up working. 

Related to this point, Jess, McMillan, and Carse [53] go on to identify three related factors that can help create a framework to bring stakeholders together, support the development of a more educational and coherent PE more closely connected to health, and thereby move the PE and health agenda forward. Each of these factors have already been touched upon but are coined here as personal vision(s), boundary-crossing, and contextualised professional development [53]. Personal visions refer to PE teachers ensuring they have a voice in debates about the future positioning of PE and share their visions for the subject with other stakeholders in terms of its relationship with education, sport, and health. Boundary-crossing refers to inter- and cross-disciplinary working and learning and breaking down boundaries between subjects to share visions, developing positive relationships with other stakeholders, and facilitating more holistic and integrated approaches to education, PE, and health [53,66]. Engaging in boundary-crossing will require the PE profession to be open-minded and outwardly facing and to move beyond traditional subject-specific thinking and many of the working practices and approaches that have tended to prevail. O’Connor and Jess [66] present some useful guiding principles as a frame to support boundary-crossing that involve engaging different stakeholders (from, in this case, different subjects and across PE, sport, and health contexts), likely with different starting points, in a participative process. It is recognised that this process is likely to be emotional and take time to evolve and develop, and that consensus may not always be possible or even necessary [53]. Time, space, and opportunities, therefore, need to be created for this through a structured and supportive process which should enhance learning and understanding between and have mutual benefits for all parties. In turn, contextualised professional development serves as an important means of facilitating boundary-crossing and encouraging PE teachers to develop, adapt, or imagine alternative visions for PE over time. Specifically, it involves long-term, continuous, collaborative, and contextualised learning, which provides opportunities for PE teachers to come together, critically engage with PE and health debates, and reflect on and adapt their practice in a situated manner. According to Jess and colleagues [53], there are many possibilities for this contextualised learning through approaches such as professional learning communities, action research, practitioner inquiry, self-study, lesson study, peer coaching, and so on.

## 5. Conclusions

In this paper, I have argued the case for PE to be at the very centre of and driving schools’ efforts to promote physical activity. Whilst the complexity of health in PE is recognised, and the subject faces some challenges that raise questions concerning the responsibility and effectiveness of PE in promoting physical activity, there have been some very positive strides in recent years that serve to support, highlight, strengthen, and reinforce the focus and responsibility of PE in this endeavour. In light of these, the next few years will be a pivotal time for PE. Furthermore, there have been many new and positive developments which, if embraced, will help PE to realise its potential with respect to the promotion of physically active lifestyles moving forwards. High-quality PE that has young people at the core, and provides positive and meaningful physical activity and learning experiences, is clearly critical and should provide the bedrock and serve as the springboard to wider efforts and a whole school commitment to promoting physical activity. Achieving this may demand some upskilling of teachers, challenging some longstanding traditions and practices, and sharing an extended and holistic vision for the subject such that it truly meets the needs of our youth of today. It is both time and timely for the PE profession to be bold, have confidence, grasp the opportunities, and ensure that high-quality PE is central to the explicit planning and co-ordination of meaningful, coherent, relevant, and sustainable physical activity opportunities for young people in schools. In this way, more young people will likely be engaged and empowered to enjoy and reap the benefits of a physically active lifestyle.

## Data Availability

No new data were created or analyzed in this study. Data sharing is not applicable to this article.

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
