# Peer review of "Physical Education: At the Centre of Physical Activity Promotion in Schools"

_ijerph, 2023, doi:10.3390/ijerph20116033_

Round 1

Author Response

Reviewer’s Feedback

Author Response

This opinion paper captures the key issues related to the positioning of physical activity within school PE and across the whole school. The paper takes both a detailed and balanced approach to the topic. The argument is well-constructed and follows a clear line of development.

I would be happy for the paper to be published in its current form, but have a number of minor points for the author to consider.

1. The author correctly highlights the problematic relationship that sport and physical activity have long had with PE. This issue is partly addressed, but it may be worth considering how the boundary crossing noted towards the end of the paper could be employed to articulate a more integrated form of PE in which sport, physical activity and health are more closely aligned. The author may wish to give more detailed thought as to how this may be enacted in the future.

2. The vast majority of the examples presented throughout the paper are from English speaking countries. Given the international readership of the journal, could the author include more examples from other countries, particularly non-English speaking countries?

3. Given the multi-national nature of the UK, the paper tends to conflate what is happening in England with the rest of UK (although this division is acknowledged later in the paper). It would be helpful, perhaps as a footnote, to highlight that the education systems of England, Northern Ireland, Scotland and Wales, are all devolved and that the PE policies in these countries are quite different.

As such, the recent work ‘core subject’ debate which is covered in some detail only refers to what is happening England.

4. As a PE teacher educator, the author may wish to offer some opinion about the way that teacher educators could work to ensure the physical activity and health are included as part of the initial teacher education and continuing professional development of teachers and other practitioners.

5. A number of sentences, particularly at the beginning of sections, are long. As a reader, it would be helpful if these sentences were split into two or even more sentences e.g. Lines 101-105, 235. I was unsure if the double use of whole school in the first sentence of the abstract was necessary.

6. The referencing will need to be amended to follow the journal guidance.

Overall, this is a clear and insightful paper that offers a detailed and balanced view of physical activity and health within school PE.

Many thanks for your positive feedback on my manuscript.  I really appreciate you taking the time to review it and I’m pleased you feel the paper is detailed, balanced and that the argument is well constructed and clear. 

This is welcomed news, but many thanks for the helpful suggestions which I have tried to address in this re-submission.

This is a good point.  I have further explored boundary crossing, the principles it will likely involve (with reference to recent literature on boundary crossing), and how it might be employed in the future towards the end of the paper (please see pages 15-16). 

This is a fair observation. I have now extended the examples in the manuscript to cite non-English speaking countries, including Sweden and Norway and acknowledged these examples represent just a few (see page 2).

Apologies for not making this clear in the paper.  I have explained the devolved government and different education systems and policies in England, Northern Ireland, Scotland and Wales as a footnote as suggested (see page 7).  I have also identified the four UK countries in the text and provided a little more detail about the curricula developments in Northern Ireland, Scotland and Wales (see page 14).

I have also made it clear in the text that the ‘core subject’ debate relates to the National Curriculum in England (see pages 7 and 8).

Many thanks for suggesting this.  I have referred to the role of teacher educators and continuing professional development in supporting the effective promotion of physical activity in PE (e.g. by addressing and upskilling trainee and practising teachers in the area of health). Please see page 11.

I have read through the paper again and shortened some of the longer sentences, particularly in the opening to sections.  I have also removed one of the references to ‘whole’ school in the first sentence of the abstract.

I have amended the references in accordance with the format for the journal now.

Many thanks once again for your positive comments and for your really helpful suggestions.

Reviewer 2 Report

The paper focuses on the meaningful provision of physical activity for children and young people, aiming for social, emotional, mental, as well as physical wellbeing in the long term. The author's opinion is based on a purposeful synthesis of the analysis of documents from leading world organizations and recent scientific publications. Currently, in many countries, schools focus on academic achievement, but areas such as emotional and physical development are not a priority. Although this paper expresses a theoretical opinion, it makes us think about the development of the individual as a whole. Thanks to the author for this work!

Reviewer 3 Report

I am sorry that I have to be critical to this manuscript. The what the authors argue is not really very interesting and does not maintain scientific methods. The authors argue "PE to be at the very heart of and driving schools’ efforts to promote physical activity." This is the same thing as arguing that math education has to be at the heart of math activities. I really do not see a point of this article. 

Author Response

Reviewer’s Feedback

Author Response

I am sorry that I have to be critical to this manuscript.

What the authors argue is not really very interesting and does not maintain scientific methods.

The authors argue "PE to be at the very heart of and driving schools’ efforts to promote physical activity." This is the same thing as arguing that math education has to be at the heart of math activities. I really do not see a point of this article.

Please don’t apologise for your critique.  I really appreciate you taking the time to review the manuscript and provide your view on it.  The process and your comments have encouraged me to further reflect on and explain/clarify the messages conveyed in the article and the reasons underpinning it.  I therefore hope the manuscript and its intentions are now clearer as a result. 

This is an ‘opinion piece’ and therefore does not adopt scientific methods in an empirical research sense.  However, I have tried to draw on a good deal of literature from various sources (and integrated further references into this re-submission) to ensure my arguments are evidence-informed.  I have also revisited and extended some parts of the discussion in an effort to add interest for and engage readers.

I agree that this argument may seem to be stating the obvious.  However, as the manuscript goes on to explain in the ‘hindering PE’s potential’ section, PE faces a number of longstanding challenges with regards the promotion of physical activity and there are different schools of thought on the matter and on the role of PE in physical activity promotion.  Some authors/researchers for example, question whether the subject should/does have a role to play at all.  Physical activity behaviour and promoting physical activity are complex with the latter representing much more than providing physical activity opportunities for students through different PE activities and expecting this approach to be effective.  I have acknowledged your point (about PE being at the heart of schools’ efforts to promote physical activity seeming obvious) (see page 3), and I can certainly see from where this viewpoint derives.  In addition, I have developed the argument/discussion in a number of places to explain the challenges, tensions and complexity of PE’s role in physical activity promotion in more detail, and hence try to better clarify why PE may not be so central to or effective in this regard in practice (see pages 4, 8, 10, 11).  Overall, I therefore hope the point of the manuscript is now clear.